# Highly Strong and Damage-Resistant Natural Rubber Membrane via Self-Assembly and Construction of Double Network

**DOI:** 10.3390/membranes12100933

**Published:** 2022-09-26

**Authors:** Heliang Wang, Fanrong Meng, Mingyuan Yi, Lin Fang, Zhifen Wang, Shoujuan Wang

**Affiliations:** 1State Key Laboratory of Biobased Material and Green Papermaking, Qilu University of Technology (Shandong Academy of Sciences), Jinan 250353, China; 2College of Materials Science and Engineering, Hainan University, Haikou 570228, China

**Keywords:** natural rubber latex membrane, self-assembly, double network, toughening, sodium lignosulfonate

## Abstract

Natural rubber latex (NRL) is commonly employed to manufacture medical protective appliances. However, the characteristics of weakness and fragility of NRL membranes limit their further application. To achieve excellent strength and damage-resistance of the rubber membrane, this work reported a facile core–shell structure construction strategy via self-assembly with modified sodium lignosulfonate (MSLS) and NRL to create a tough membrane. The double network can be formed after introducing polyamide epichlorohydrin resin (PAE) into the NRL membrane. Specifically, the first robust MSLS-PAE network can break in advance to dissipate applied energy, thereby achieving high fracture energy and tensile strength of ~111.51 kJ m^−2^ and ~37 MPa, respectively, which overtakes numerous soft materials. This work facilitates more studies on latex/lignin-based products with high performance and good stability for the functional application of biopolymer.

## 1. Introduction

Natural rubber latex (NRL) is commonly used in the manufacture of medical protective equipment and daily necessities, which are required to possess excellent strength and stretchability. However, the weakness and fragility of NRL membranes limit their extended application [1,2,3]. Currently, the major toughening NRL membrane method generally includes the preparation of nanofiller suspension and in situ polymerization of the stiff resin. The in situ polymerization of resin in latex can achieve a better reinforcement effect. For example, styrene is in situ polymerized in natural latex under the action of an initiator, thereby improving the mechanical properties of the material [4]. However, there are still limitations in practice due to the economic benefits and environmental protection issues. Correspondingly, an alternative approach to synthesizing a suitable nanofiller suspension has been extensively adopted [5]. Indeed, some filler suspensions can enhance the tensile strength of the product, whereas the even dispersion of fillers remains a challenge since the fillers can affect the stability of NRL particles [6]. Consequently, the elongation at break cannot be well maintained [7].

Therefore, seeking new materials and techniques is one of the means to prepare considerably tough and stretchable NRL membranes, and to achieve the pursuit of green and sustainable rubber latex products in the modern rubber industry. In view of these situations, lignin, as the second most abundant biomass and the only aromatic biopolymer in plants, has been deemed as an eco-friendly reinforcing filler [8,9,10,11,12,13]. However, the direct introduction of raw lignin into rubber latex via conventional technologies barely helps with the enhancement of mechanical performance. A high percentage of lignin (25 phr) is required to improve mechanical properties [14,15,16]. This is due to the poor compatibility between rubber chains and lignin particles. Specifically, the high content of hydroxyl groups on the lignin and the voids around the lignin aggregates resulted in poor interfacial adhesion and compatibility between lignin and rubber chains [17,18].

To circumvent the above issues, the experiment constructed a double-crosslinked network assisted by modified sodium lignosulfonate (MSLS) and polyamide epichlorohydrin (PAE), which was well dispersed in the natural rubber latex matrix to restrict the motion of fillers and avoid the separation of fillers with the rubber matrix. The filler in the rubber matrix migrated to the surface with prolonged storage time. At the same time, the formation of the MSLS-PAE network limited the movement of the filler to a certain extent since the MSLS was chemically crosslinked by PAE. Moreover, during the preparation of rubber film, two stages of pre-vulcanization and vulcanization were conducted. The pre-vulcanization stages were mainly the reaction of MSLS and PAE and the partial vulcanization of the latex particles. The vulcanization stage was used to further vulcanize the dried and pre-vulcanized rubber film, which mainly aimed to fully vulcanize the natural rubber with sulfur to form a crosslinking network. The well-designed strategy concurrently achieved high breaking elongation, strength, and fracture toughness. In detail, the MSLS is adsorbed to the surface of the rubber particles after self-assembly caused by electrostatic (electric double layer effect) and π–π interactions, leading to the formation of a core–shell-like structure. Subsequently, PAE worked as a bridge to connect -COOH groups on the MSLS to construct a stiff MSLS-PAE network that interspersed in the rubber matrix during the pre-vulcanization of the natural rubber latex (Figure 1a,b). In addition, the latex particles were also crosslinked under the influence of the vulcanization accelerator, and the formed emulsion was further dried and vulcanized to form a double network membrane. Therefore, when exposed to externally applied force, the MSLS-PAE network preferentially breaks and works as a stiff network to dissipate energy to improve the strength and toughness of the membrane.

## 2. Materials and Methods

### 2.1. Materials

The raw material information used in the experiment is shown in the Table 1, below. 

### 2.2. Sample Preparation

#### 2.2.1. Modification of Sodium Lignosulfonate

A total of 50 mL of water and 20 g of sodium lignosulfonate (Table 1) were added to the beaker and then stirred to dissolve. Thereafter, 10 mL of hydrogen peroxide was dropped into above solution at 80 °C for 2 h to obtain carboxylated sodium lignosulfonate. Subsequently, HCl was used to adjust the pH (<2) to obtain a precipitate product. After filtering and drying, the final yield was about 10%. The carboxyl content of MSLS measured by titration method with 0.19 mmol/g.

#### 2.2.2. Pretreatment of PAE

Calcium carbonate was used to remove dilute sulfuric acid in PAE to obtain calcium sulfate precipitation, and the purified PAE was acquired after layering.

#### 2.2.3. Preparation of NRL Vulcanized Membrane

The natural rubber latex with a solid content of 100 g (the NRL concentration was 98%: 0.52% ammonia and 60.65% solids) was added with a compounding agent: the solid content of 45% sulfur, 55% ZDC, 40% zinc oxide, and 20% KOH were 0.75 g, 0.75 g, 0.4 g, and 0.3 g, respectively. Finally, 16 mL of water was added to the mixture.

Preparation of MSLS/NRL: dissolved 0.75 g, 1.0 g, 1.25 g, 1.5 g, and 1.75 g of modified sodium lignosulfonate in 15 mL of distilled water to adjust the pH of the solution to 10.5 to form MSLS suspensions of different concentrations. Then, added NRL to the above to form MSLS/NRL.

Preparation of MSLS-PAE/NRL: on the basis of the optimum content of MSLS/NRL (determined by tensile strength and tear strength), added different contents of 12.5% PAE (1.6 g, 3.2 g, 4.8 g, and 6.4 g) to prepare MSLS-PAE/NRL.

The above three emulsions were stirred in a mixer at 35 rpm for 30 min. Then pre-vulcanized the emulsion in a water bath for one hour (the temperature slowly changed from 25 °C to 60 °C within 30 min, and then pre-vulcanized for 30 min after reaching 60 °C). Placed the pre-vulcanized emulsion on a horizontal glass plate to dry naturally, and then soaked the dry membrane in water at 60 °C for 10 min to clean the surface residues. Finally, it was put into an oven at 100 °C for vulcanization for 2 h and 15 min to obtain vulcanized NRL membrane, MSLS/NRL membrane, and MSLS-PAE/NRL membrane.

### 2.3. Characterizations

The mechanical properties of the samples were measured by GOTECH AI-3000 testing machine. The tensile speed was 500 mm/min, and the sample was a thin, dumbbell-shaped rod with a center size of 20 mm × 4 mm × 1 mm. The cyclic tensile test was performed at a strain rate of 100 mm/min. The specimen was a thin, dumbbell-shaped strip with central dimensions of 20 mm × 4 mm × 1 mm. For every cycle, the dissipation energy (*U*) was calculated as follows [18]:(1)$$U=\int_{\mathrm{loading}}^{\;}\sigma d\lambda -\int_{\mathrm{unloading}}^{\;}\sigma d\lambda$$
where *σ* is the stress, and *λ* is the strain.

The fracture energy test was obtained by testing machine (GOTECH AI-3000, Taiwan, China). The size of sample was 1 mm × 5 mm × 10 mm, and the notch was about 2 mm in the middle of the rectangular specimen. The classic single-edge method was used to determine the fracture energy (*G_C_*), and the calculation process was according to the following formula [19,20,21]:(2)$$G_{C}=\frac{6W_{C}}{\sqrt{\lambda_{C}}}$$
where *c* is the notch length, *λ_C_* is the strain when the sample breaks, and *W* is the strain energy density calculated by integrating the stress–strain curve of the unnotched sample until *λ_C_* − 1.

Fourier transform infrared (ATR-FTIR) spectroscopy (Bruker Vertex 70, Bremen, Germany) was used to determine the samples before and after the reaction. In order to study the vulcanization kinetics of rubber membrane, vulcanizer (GOTECH M-3000 AU, Taiwan, China) was used to characterize the vulcanization curve of the samples at 100 °C. The crosslink density of vulcanized NRL composites in cyclohexane was determined by using an equilibrium swelling method and calculated by the classical Flory–Rehner equation [22].

To further study the core–shell formation mechanism between MSLS and rubber, the interaction energy (Δ*E*_interaction_) was investigated by molecular dynamics simulation. DFT calculation was conducted at ωB97X-D/6-31g** using G09 Software (Gaussian Inc.,Wallingford, CT, USA) [23,24]. Consequently, the interaction energy was calculated from the following equation [25]:$$\Delta E_{interaction}=\left(EAB\;-\;\left(EA+EB\right)\right)$$
where *E*_AB_ and *E*_A_ (and *E*_B_) are the total potential energies of final system and initial system, respectively.

Note: structures A and B were used to represent MSLS and natural rubber segment, respectively.

In a nitrogen atmosphere, on the TA dynamic mechanical analyzer (DMA 850, TA Instruments, New Castle, DE, USA) with a gas cooling accessory, the temperature-dependent curve of the loss factor (tanδ) in the tensile mode was obtained. In all cases, a preload force of 0.01 N was applied. Differential scanning calorimeter (DSC): in a nitrogen atmosphere, the heat flow curve of the sample was obtained on the instrument (TAQ100, TA Instruments, New Castle, DE, USA). The heating rate was 10 °C/min.

Scanning electron microscopy (SEM) observed the morphology and the element distribution of latex particles (on Verios G4 UC, ThermoFisher Scientific, Waltham, MA, USA) in order to understand the construction of double network realized by MSLS-PAE during the NRL pre-vulcanization process. This work avoided the interference caused by the addition of compounding agents, and MSLS and MSLS-PAE were added to the NRL without the compounding agents for pre-vulcanization. The morphology of latex particles and formation of crosslinking, therefore, were observed by transmission electron microscopy (TEM) at an accelerating voltage of 200 kV (FEI Talos F200C, ThermoFisher Scientific, Waltham, MA, USA). MSLS suspension of the particle size distribution and electric potential were measured by the laser particle sizer (Zetasizer nano ZS 900, Malvern Panalytical, Malvern, UK.), and the concentration of latex was diluted to about 0.02% of its original state. pH of the solution after dilution was 7.3.

Determination of mechanical stability of latex: the mechanical stability of the latex was determined according to the standard of ISO 35 (natural rubber latex concentrate—determination of mechanical stability). Moreover, the time from the start of stirring to the appearance of small curds in the latex was regarded as the time to mechanical stability.

Determination of latex viscosity: the viscosity of the pre-vulcanized latex was tested with a falling ball viscometer (VISCOBALL, Fungilab, Barcelona, Spain). Under certain temperature conditions, measured the time that the test ball rolled and fell in a measuring tube filled with test solution with a certain inclination angle of 10°, and calculated the dynamic viscosity of the test solution according to the following formula, in mPa·s:$$\eta =k\left(\rho -\rho 1\right)t$$
where *η* (mPa·s) is the dynamic viscosity of the test solution, *k* (mPa·s/(g·cm^−3^·s)) is test ball constant, *ρ* (g/cm^3^) is density of test ball, *ρ*1 (g/cm^3^) is density of test solution (0.95 for concentrated natural rubber latex), and *t* (s) is the average time for the ball to fall.

## 3. Results and Discussion

Self-assembly refers to the spontaneous association of stem individuals at the same time and aggregation to form a tight and orderly structure, which is a complex synergy of the whole [26]. To investigate the self-assembly process and construction of the crosslinking network, TEM was, as a consequence, applied to observe the process. For better observation, the work simulated the pre-vulcanization process, in which no vulcanization accelerator was added to avoid interference caused by other components. As displayed in Figure 2, the latex particles of pure NRL exhibited a relatively regular round structure (Figure 2a), while the surface of the NRL latex particles with the MSLS displayed uneven geometric shapes (Figure 2b), indicating that the MSLS was absorbed into the NRL particles after self-assembly. Since sodium lignosulfonate can be used as an anionic surfactant and there is a π–π interaction between the benzene ring (on MSLS) and a double bond (on NR), we speculated that MSLS could be adhered to the surface of latex particles to increase the particle size under the above interactions [27,28]. After introducing PAE into the NRL-MSLS system, the previously obtained latex particles were bridged together. Therefore, during the simulated pre-vulcanization heating process, PAE may crosslink the MSLS on the surface of the NRL particles, thereby constructing the crosslinked morphology of the latex particles and eventually forming a double network with the rubber matrix (Figure 2c). In order to further prove this point of view, the element distribution of the above samples was observed by SEM (Figure 2d and Appendix A), demonstrating that S, O, and N elements are well dispersed on the surface of MSLS-PAE/NRL. This confirmed that the core–shell structure of latex particles was structured by MSLS after self-assembly, and PAE can work as a bridge to connect each colloidal particle. To theoretically investigate the formation process of self-assembly between MSLS and NRL, DFT simulation (Figure 2e and Appendix A) was then performed to quantify the interaction of MSLS and rubber chains. The obtained result demonstrated the interaction among rubber chains and MSLS values at ~41.73 kcal/mol (Appendix A), suggesting that a relatively powerful interaction can be reached between MSLS and rubber, which may result from the π–π interaction between the double bond on rubber and the benzene ring on MSLS [29]. Moreover, after the optimization of the simulated structure on the level of ωB97X-D/6-31g**, the electrostatic potential map was drawn by the Surface and Cubes function module of GaussView software. The blue part represents areas of high positive charge density, while the red and yellow parts represent the areas that bring a negative charge. A strong interaction can be achieved among blue and red/yellow parts. It is found from the obtained molecular electrostatic potential image (Figure 2e) that the benzene ring exhibits relatively positive charges due to the strong electronic attraction of the -COOH and -SO_3_ groups on its two sides. The double bond is relatively electron deficient compared to the benzene ring, the electrostatic interaction therefore may be generated among the benzene ring and 1-methyl-2-pentene, and the simulated MSLS structure may wrap around the 1-methyl-2-pentene, leading to the self-assembly of the MSLS and NRL and the formation of the core–shell structure.

Thereafter, the change of particle size and Zeta potential for the above process were then conducted by dynamic light scattering test (DLS), shown in Appendix A and Appendix A. It was found that the potential for NRL was increased after the attachment of MSLS. Since the reaction between PAE and MSLS lowers the Zeta potential, the potential of the final system, as expected, decreased. In terms of particle sizes, the average particle size gradually became larger. Therefore, the above results demonstrated the formation of a crosslinking network between PAE and MSLS.

The construction of a double network originated from modified sodium lignosulfonate (MSLS) and PAE. Thus, the group changes before and after modification of sodium lignosulfonate were studied by FTIR, shown in Figure 2f. In the spectrum of SLS, the strong CH bands at 2927 and 2848 cm^−1^ correspond to methoxy and hydroxymethyl groups, and 3392 cm^−1^ corresponds to hydroxyl (-OH) [30]. On the other hand, the peak intensity of 1563 cm^−1^ can be attributed to the symmetrical aromatic skeleton vibration indicated by the lignin macromolecule. The peak is about 1328 cm^−1^ because syringyl (S) has a free C5 position in the aromatic ring. The CO stretching vibration bands of the primary and secondary ethanol and ether are at 1039 cm^−1^, corresponding to the aromatic CH of the G unit at carbon positions 2, 5, and 6 at 825 cm^−1^ out-of-plane deformations and vibration [31]. After oxidation by hydrogen peroxide, strong characteristic peaks of carboxyl groups (-COOH) appeared at 1714 and 1652 cm^−1^, indicating that lignin was successfully oxidized to produce carboxyl groups. Meanwhile, during the oxidation, a dark insoluble component precipitated from the solution, suggesting that the hydrophilicity of sodium lignosulfonate was dramatically changed (Appendix A). As shown in Appendix A, the hydrophobicity was improved after the modification, in which the water contact angle was enhanced from 51.78° to 108.65°. Moreover, the particle size in the MSLS suspension was also measured, and the particle size distribution was found to be concentrated around 50–400 nm, as shown in Appendix A.

To figure out the enhanced performance that was brought by the MSLS and double network, the appropriate amount of MSLS in NRL was determined through mechanical performance testing in advance. The dosage of MSLS ranging from 0.75 phr (per hundred rubber) to 1.75 phr was used to explore its tensile strength and tear strength. As shown in Appendix A, the tensile strength and tear strength of NRL increased along with the MSLS contents, in which the maximum value was reached when the content was 1.5 phr while the corresponding tensile strength and tear strength were 32.33 MPa and 48.37 KN m^−1^, respectively (Appendix A). Compared with NRL without MSLS, a significant enhancement in mechanical properties was realized, indicating that carboxylated lignin had a reinforcing effect in NRL since it can work as a considerable filler for polymer matrix, and the compatibility with rubber matrix may also be enhanced, as confirmed by the above water contact angle test [32,33,34]. Subsequently, PAE was added under a fixed MSLS content of 1.5 phr to construct a stiff and double network with MSLS for the further improvement of mechanical properties [35]. As shown in Figure 3a,b, the maximum tensile stress increased with the content of PAE, and the elongation at break, tensile strength, and tearing strength reached the maximum when the content was 4.8 phr, valuing at ~880%, ~37.1 MPa, and ~61.08 kN m^−1^, respectively. Compared with NRL (elongation at break 780%, tensile strength 25.37 MPa, and tear strength 41.39 kN m^−1^), the values have been increased by 12.8%, 46.23%, and 47.57%, respectively.

In order to explain the superior mechanical properties of MSLS-4.8PAE, the experiments were analyzed from the perspective of energy consumption from cyclic loading tests, shown in Appendix A. It can be found that the hysteresis loop of NRL was smaller than that of MSLS-PAE, and the hysteresis loop of MSLS-PAE varied with the content of PAE. By calculating the integral area of each hysteresis loop, as shown in Figure 3c, the energy dissipation of NRL was ~4.65 MJ m^−3^ at 600% strain, while that for MSLS-4.8PAE reached the maximum energy dissipation, valuing at ~6.82 MJ m^−3^. This may be attributed to the MSLS-PAE crosslinked network working as the first plastic network to preferentially break, leading to the consumption of applied energy and the decline in the rupturing of elastic rubber chains (Figure 3d) [50]. Reported studies show that soft materials containing weak and strong networks can be toughened by sacrificial fracture of weak networks [51,52]. Since the MSLS-PAE worked as an efficient sacrificial network in the matrix and could dissipate much energy rapidly and effectively under deformation, the material was tested for damage resistance characterized by fracture energy [53]. Benefiting from a stiff network constructed by PAE and MSLS, the fracture energy, as expected, reached up to ~111.51 kJ m^−2^ (Figure 3e) under the optimal dosage of MSLS and PAE, which overtakes toughened nature rubber (~50 kJ m^−2^) and many synthetic robust soft materials (generally less than 100 kJ m^−2^, Figure 3f).

To explore the underlying mechanism of their excellent mechanical properties, the vulcanization curve and crosslink density of various samples were investigated. For better observation, the samples used to test the vulcanization characteristic curve were dried directly into a membrane without pre-vulcanization. The maximum torque value increased with the percentage of PAE, in which the maximum torque value was achieved at 4.8 phr, displaying a similar result to that for mechanical properties (Figure 4a). Since the torque value, to a certain extent, reflects the degree of crosslinking of the material [54], the development in the torque value increases the degree of crosslinking [55], which is likely to be related to the addition of PAE content. Subsequently, the crosslinking density, an indirect parameter of mechanical properties, was measured for the vulcanized and crosslinked membranes [56]. As shown in Figure 4b, the crosslink density of MSLS-4.8 PAE, as expected, was significantly higher than that of NRL and other PAE addition systems, which was in good agreement with the results of vulcanization characteristic curves. The increase in the crosslinking density of MSLS-PAE is probably due to the crosslinking of PAE and MSLS to form a new crosslinked network. In addition, to meet the realistic demands, the stability (including thermal and mechanical properties) was further studied. The thermal behaviors and influence of a stiff network for the above samples were firstly investigated (Figure 4c) since elastic nature is a key point for realistic application for related soft products. As expected, the glass transition temperature (Tg) of MSLS-PAE/NRL was lightly improved (~3 °C improvement, Figure 4c) compared to that of the NRL system from dynamic mechanical analyzer (DMA) curves, which was also confirmed by differential scanning calorimeter (DSC) results (Figure 4d), illustrating that the introduction of a PAE stiff network had not greatly affected the motion of elastic molecular segments and may not influence the usage of related products [57,58,59,60].

Apart from that, as a milky industrial raw material, natural rubber latex must possess sufficient colloidal stability to withstand the mechanical effects of stirring, centrifugation, and transportation. In practice, the mechanical stabilization time of latex is an important parameter to characterize the stability of concentrated natural rubber latex. This stability is essential for the transportation and storage of NRL before the product is formed. Many studies have shown that adding different types of surfactants or carboxy-based soaps can increase the mechanical stabilization time value of NRL and can adhere to the surface of NRL particles. MSLS, as a surfactant that contains carboxyl groups, increases the mechanical stabilization time value compared with NRL (Figure 4e). This also proves that the increase in the particle size of the colloidal particles (Figure 4f) is caused by the attachment of MSLS to the surface of the NRL particles. The viscosity of NRL increases after adding MSLS-PAE, which is due to the thickening effect caused by the addition of PAE. However, a slight increase in viscosity does not affect its colloidal properties. This shows that MSLS-PAE/NRL can be applied in practice.

## 4. Conclusions

In summary, a strong, tough natural rubber latex membrane was fabricated by MSLS and NRL through self-assembly and the formation of a core–shell structure, in which PAE worked as a bridge to connect MSLS particles to constrict double networks with a rubber matrix. To confirm the formation process, TEM, Zeta potential, particle size distribution, and DFT simulation were performed in this work to prove the core–shell construction mechanism between PAE and MSLS. In detail, the adsorption of MSLS on the surface of natural latex particles is mainly achieved by electrostatic interaction between the benzene ring (on MSLS) and 1-methyl-2-pentene (on NRL). Subsequently, Fourier transform infrared spectroscopy confirmed that MSLS and PAE formed a crosslinked network structure. The mechanical properties of the MSLS-PAE/NRL membrane were adjusted by the dosages of MSLS and PAE, and the best mechanical properties were achieved when the dosages of MSLS and PAE were 1.5 and 4.8 phr, respectively. Compared with the pure NRL membrane, the elongation at break, tensile strength, and tear strength of the MSLS-PAE/NRL membrane were increased by 12.8%, 46.23%, and 47.57%, respectively. Meanwhile, the fracture energy of the MSLS-PAE/NRL membrane reached ~111.51 kJ m^−2^, which overtakes numerous soft materials. Notably, the introduction of MSLS and PAE stabilizes the NRL instead of affecting its stability.

## Figures and Tables

**Figure 1 membranes-12-00933-f001:**
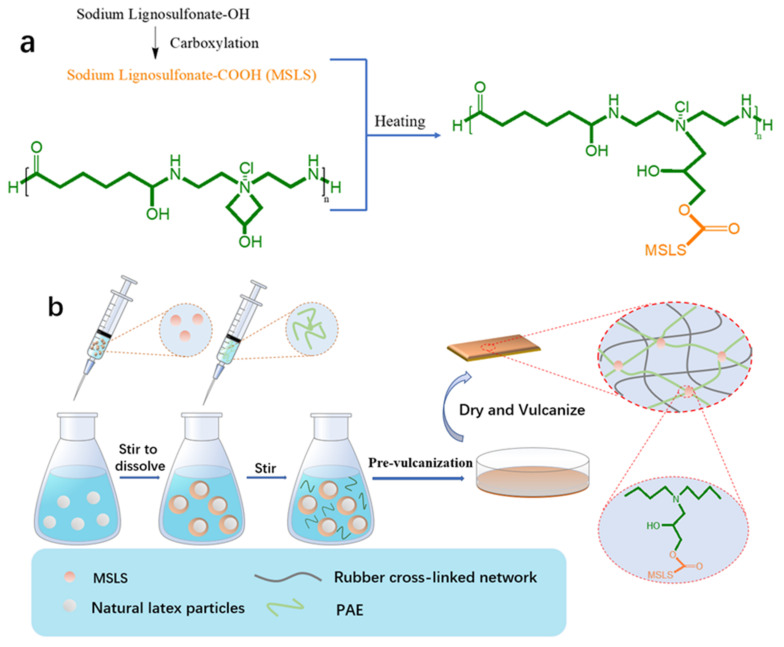
(**a**) Potential MSLS-PAE crosslinking mechanism, and (**b**) schematic illustration for MSLS-PAE/NRL double network construction.

**Figure 2 membranes-12-00933-f002:**
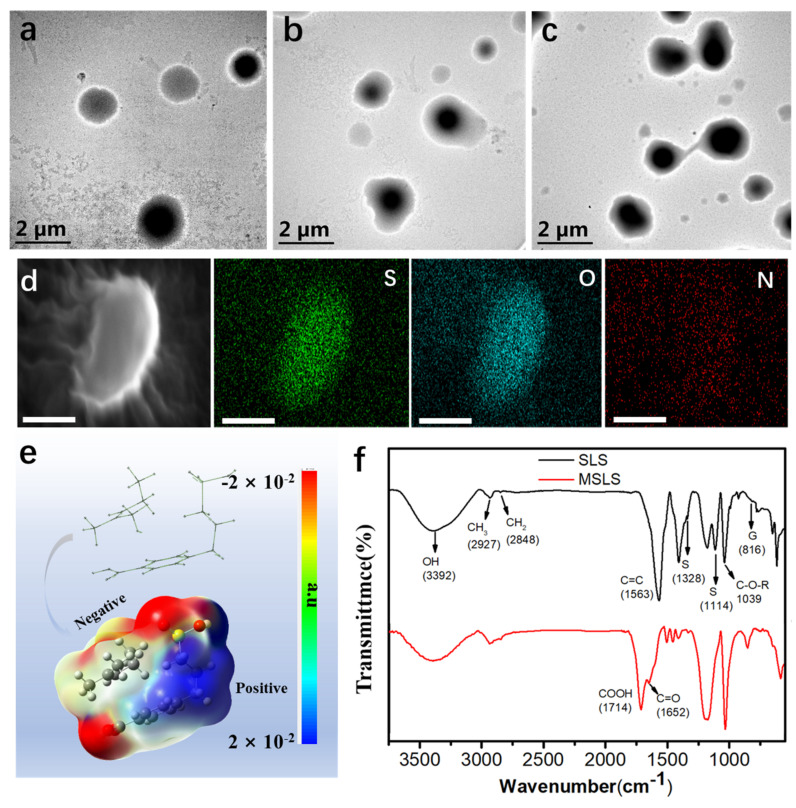
TEM images of simulated pre-vulcanized NRL. (**a**) NRL, (**b**) MSLS/NRL, (**c**) MSLS-PAE/NRL, (**d**) SEM image of element distribution (S, O, and N) of MSLS-PAE/NRL particles (the scales bar is 2 μm), (**e**) molecular electrostatic potential image, and (**f**) RTIR spectra of sodium lignosulfonate (SLS) and MSLS.

**Figure 3 membranes-12-00933-f003:**
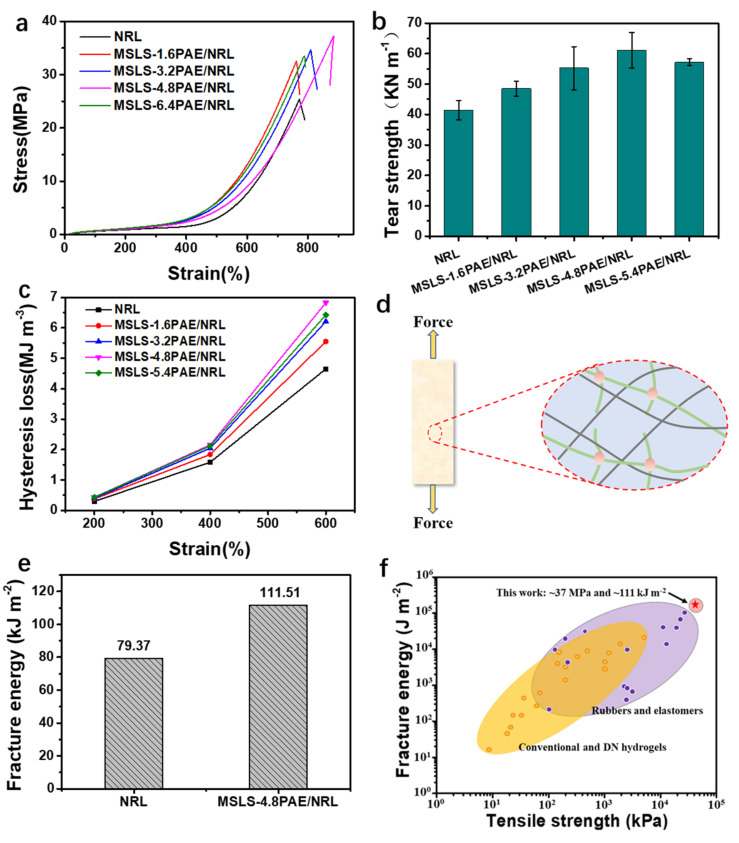
(**a**) Tensile strength, (**b**) tear strength, (**c**) the hysteresis loss of different materials under different strains, (**d**) the potential schematic illustration of the microstructure changes in MSLS-PAE-functionalized rubber matrix during stretching, (**e**) fracture energy of MSLS and MSLS-PAE/NRL, and (**f**) comparison of tensile strength and fracture energy of the obtained materials with other soft materials [35,36,37,38,39,40,41,42,43,44,45,46,47,48,49].

**Figure 4 membranes-12-00933-f004:**
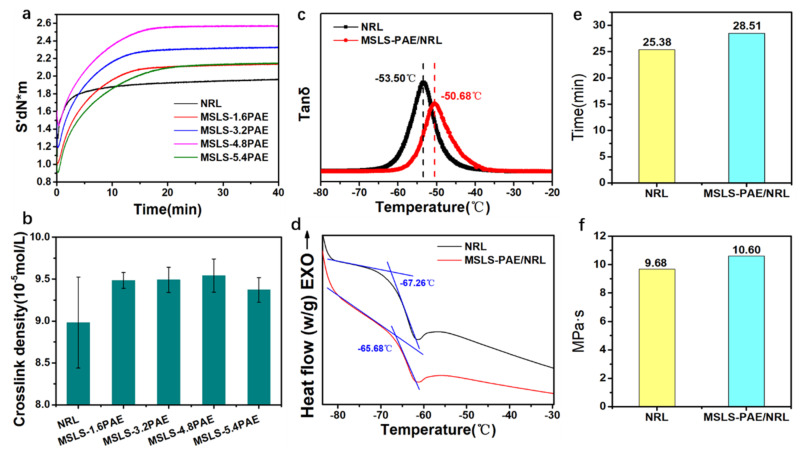
(**a**) Vulcanization characteristic curve, (**b**) crosslink density of natural rubber latex with different PAE content, (**c**) DMA and (**d**) DSC curves of NRL and MSLS-PAE/NRL, (**e**) mechanical stabilization time, and (**f**) viscosity of NRL and MSLS.

**Table 1 membranes-12-00933-t001:** Material information.

Sample Name	Specification	Source
Sodium lignosulfonate	Tech	Aladdin Chemical Co., Ltd., Shanghai, China
Hydrogen peroxide (30%)	AR	Xilong Science Co., Ltd., Shantou, China
HCl	CP	Guangzhou Chemical Reagent Factory., Guangzhou, China
NaOH	AR	Xilong Science Co., Ltd., China
Calcium carbonate	AR	Tianjin Fuchen Chemical Reagent Factory., Tianjin, China
KOH	AR	Xilong Science Co., Ltd., China
Natural rubber latex (60%)	Tech	Hainan Jinlian Rubber Processing Branch., Haikou, China
Polyamide epichlorohydrin (PAE)	Tech	Shandong Tiancheng Chemical Co., Ltd., Yanzhou, China
Zinc oxide (ZnO)	Tech	Shanghai Mackin Biochemical Technology Co., Ltd., Shanghai, China
Zinc diethyldithiocarbamate (ZDC)	Tech	Shanghai Mackin Biochemical Technology Co., Ltd., China
Sulfur (S)	Tech	Shanghai Mackin Biochemical Technology Co., Ltd., China

## Data Availability

Not applicable.

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
