# Peer review of "Highly Strong and Damage-Resistant Natural Rubber Membrane via Self-Assembly and Construction of Double Network"

_membranes, 2022, doi:10.3390/membranes12100933_

Round 1

Reviewer 1 Report

The article is interesting for researchers in the field of rubber, according to this the article can be published but there are some aspects to improve from the article:

Do not “we” in the article, do not use these type of personal expressions

Why you refer in the abstracta to this? “The approach should be universal for a versatile lignin-based membrane”, explain and clarify the term lignin-based from the text

In figure 2: “TEM images of simulated pre-vulcanized” , what do you mean with the simulated images? Clarify this

Put a table with the information of all the compounds/samples performed

Explain the contributions of your work

Conclusions must be redone, are weak and very summarized. Explain the reason why the samples change with the presence of PAE

Author Response

Thanks for your comment, the attached reply has been uploaded.

Reviewer 2 Report

The authors prepared NR sheet with improved mechanical properties. Few points should be clarified.

1) In the introduction, the authors stated that "To circumvent the above issues, we constructed a double crosslinked filler network assisted by the modified sodium lignosulfonate (MSLS) and polyamide epichlorohydrin 48
(PAE), which was well dispersed in the natural rubber latex matrix to restrict the motion of fillers and avoid the separation of fillers with rubber matrix.". What is actually the filler in this work?
2) The specimens were prepared as 1 mm thick so it can be considered as a sheet. The authors should test their mechanical properties following some standard (e.g. ISO, ASTM)
3) The elongation at break and modulus should also be reported.
4) Statistical treatments of mechanical properties should be performed.

Author Response

(The authors gave the same response as above.)

Reviewer 3 Report

Heliang Wang describes the production of a NRL / modified sodium lignosulfonate compound crosslinked by a charged water-soluble aliphatic polyimide and its mechanical characterization. The topic is of scientific interest, although necessary improvements are required. The English expression must be carefully checked through the manuscript.

 The introduction could be improved and extended with relevant literature. Referring to the introduction, please improve the following points.

 The author should explain more carefully what is described as “self-assembly” and provide other examples of NRL/filler systems where this effect was observed. If relevant, please consider citing:

 Tagliaro, Irene, et al. "The self-assembly of sepiolite and silica fillers for advanced rubber materials: The role of collaborative filler network." Applied Clay Science 218 (2022): 106383. https://doi.org/10.1016/j.clay.2021.106383.

 When referring to in-situ polymerization (Lines 29 – 32) please explain more in detail why it is likely chosen and provide useful references.

 Highlight the importance of NRL compounding for water soluble fillers in comparison to dry compounding and provide examples (lines 41-46).

 41- 43 it is not clear what the author means by “the incorporation of lignin into rubber latex is barely enhancement effect through conventional technologies”.

 49 – 50 what does the author mean for “restrict the motion of fillers”? Please clarify.

 Other considerations relative to the methods and the results are presented in the following lines.

 Where were “Natural rubber latex (concentration is 60%), Stearic acid (SA), zinc oxide (ZnO), Zinc di-71 ethyldithiocarbamatev (ZDC) and sulfur (S) were industrial grade products.” And HCl solution purchased? Which are the characteristics of NRL? At least define the ammonia grade and solid content.

 93 – 95 Please clarify the concentration of PAE.

 Describe the condition of analysis of Z potential. Which is the concentration and pH of the NRL suspension?

 Please make more evident when and why vulcanization or pre-vulcanization were not performed in specific observations. Please explain more clearly which is the role of each reaction and how the author is investigating those mechanisms (comment to be considered also for the introduction).

 283 Figure 8a does not exist.

 Describe how the mechanical stabilization time was assessed. There is no mention in the methods.

There is no mention of the DFT simulation in the methods. Please describe accurately the model used to describe the system.

The graphs presented in the supporting information are not described and precisely cited in the main text. Please present in the manuscript only the graphs that are discussed.

Author Response

(The authors gave the same response as above.)

Round 2

Reviewer 1 Report

There are some questions not answered by the authors, are the next: 

Explain the contributions of your work

Conclusions must be redone, are weak and very summarized. Explain the reason why the samples change with the presence of PAE

On the other hand, in line 211 "it is found from the obtained molecular electrostatic potential image (Figure 2e)" please summarize how you can define the electrostatic process and how you have simulated and graphed the Figure 2e.

Author Response

Thank you very much for your comments, the relevant documents have been uploaded.

Reviewer 2 Report

All revisions requested has been responded accordingly.

Author Response

Thanks a lot foThanks a lot for your comment.r your comment.

Reviewer 3 Report

The author satisfied all the requests of the last review.

Author Response

Thanks a lot for your comment.
